# Frontiers in integrative structural modeling of macromolecular assemblies

Kartik Majila , Shreyas Arvindekar , Muskaan Jindal and Shruthi Viswanath

National Centre for Biological Sciences, Tata Institute of Fundamental Research, Bangalore, India

## Perspective

Conformational ensembles; Electron cryo-tomography; Generative modeling; Integrative modeling; Intrinsically disordered proteins; Macromolecular assemblies; Protein language models

**Corresponding author:**
Shruthi Viswanath;
Email: shruthiv@ncbs.res.in

K.M. and S.A. authors have contributed equally.

## Abstract

Integrative modeling enables structure determination for large macromolecular assemblies by combining data from multiple experiments with theoretical and computational predictions. Recent advancements in AI-based structure prediction and cryo electron-microscopy have sparked renewed enthusiasm for integrative modeling; structures from AI-based methods can be integrated with *in situ* maps to characterize large assemblies. This approach previously allowed us and others to determine the architectures of diverse macromolecular assemblies, such as nuclear pore complexes, chromatin remodelers, and cell–cell junctions. Experimental data spanning several scales was used in these studies, ranging from high-resolution data, such as X-ray crystallography and AlphaFold structure, to low-resolution data, such as cryo-electron tomography maps and data from co-immunoprecipitation experiments. Two recurrent modeling challenges emerged across a range of studies. First, these assemblies contained significant fractions of disordered regions, necessitating the development of new methods for modeling disordered regions in the context of ordered regions. Second, methods needed to be developed to utilize the information from cryo-electron tomography, a timely challenge as structural biology is increasingly moving towards *in situ* characterization. Here, we recapitulate recent developments in the modeling of disordered proteins and the analysis of cryo-electron tomography data and highlight other opportunities for method development in the context of integrative modeling.

## Introduction

Integrative structural modeling is an approach for determining macromolecular structures that are challenging to determine experimentally (Alber et al., 2007; Sali, Glaeser, Earnest, & Baumeister, 2003). Data from multiple experiments is combined with physical principles, statistics of previous structures, and prior models for structure determination. This approach overcomes the limitations of individual techniques for structure determination and maximizes the accuracy, precision, completeness, and efficiency of structure determination (Rout & Sali, 2019; Sali, 2021).

Recent advancements in both computational and experimental domains have prompted a resurgence of interest in integrative modeling (Beck, Covino, Hänelt, & Müller-McNicoll, 2024; McCafferty et al., 2024). On the one hand, AI-based predictions of structures of proteins and their complexes with other proteins and nucleic acids have significantly advanced structural biology of late (Abramson et al., 2024; Akdel et al., 2022; Jumper et al., 2021). This has spurred the development of numerous methods that aim to integrate AI-based structures with diverse types of experimental data, including electron diffraction data from X-ray crystallography, electron density maps from electron cryo-microscopy, and chemical crosslinks from mass spectrometry (Chang et al., 2022; Stahl et al., 2024; Stahl, Graziadei, Dau, Brock, & Rappsilber, 2023; Terwilliger et al., 2022; Terwilliger et al., 2023; Zhang et al., 2023). These methods integrate the data in various ways, ranging from using the data to validate AI-based predictions, to using the data as additional inputs in the deep learning method, to encoding the data in the loss functions, resulting in structure predictions that are consistent with the data (O'Reilly et al., 2023; Stahl et al., 2023, 2024; Terwilliger et al., 2022, 2023; Zhang, Haghighatlari, et al., 2023). On the other hand, experimental techniques for *in situ* structure determination of assemblies are also rapidly advancing, with advancements in both hardware and software for imaging cells using cryo-electron tomography (Beck et al., 2024; McCafferty et al., 2024). This has led to an increase in tomography data, concurrent with an increase in the number and resolution of structures solved using tomography. Together, integrative methods using cryo-electron tomography maps along with AI-based structure predictions have resulted in significant advancements in structure determination, for example for nuclear pore complexes and ciliary complexes (Chen et al., 2023; Fontana et al., 2022; Hesketh, Mukhopadhyay, Nakamura, Toropova, & Roberts, 2022; McCafferty et al., 2024; Mosalaganti et al., 2022; Zhu et al., 2022).

Nonetheless, there is immense potential for advancing integrative modeling methods for macromolecular assemblies. Here, we provide our perspective on two areas warranting immediate method development in the context of integrative modeling: methods for modeling intrinsically disordered regions (IDRs) of proteins and approaches for leveraging *in situ* data. First, unlike ordered proteins, intrinsically disordered proteins (IDPs) comprise a dynamic ensemble of conformations that are best characterized in statistical terms rather than as static structures (Baul, Chakraborty, Mugnai, Straub, & Thirumalai, 2019). They comprise a significant fraction of the eukaryotic proteome and are involved in critical cellular processes (Oldfield & Dunker, 2014). They are found in several macromolecular assemblies, for example, the FG-Nups in the nuclear pore complex (Fontana et al., 2022; Zhu et al., 2022). However, their intrinsic disorder makes their characterization in these assemblies challenging. Improved representations for IDPs and methods for generating realistic IDP ensembles are crucial for understanding their functions. Second, the structural characterization of macromolecules using *in situ* data relies on accurate particle annotations on the tomograms (de Teresa-Trueba et al., 2023; Rice et al., 2023). However, owing to the low signal-to-noise ratio of the acquired tilt images, the missing wedge effect, and the inherent heterogeneity in the sample, the localization and identification of macromolecules in tomograms is time-consuming, laborious, and often challenging (de Teresa-Trueba et al., 2023; Moebel et al., 2021). Advances in deep learning methods and integrative approaches for combining data from other experimental and computational methods with cryo-electron tomograms can facilitate high throughput *in situ* structural characterization of macromolecular species.

In this Perspective, we first briefly review the existing integrative modeling methods and recent examples of macromolecular assemblies characterized using integrative modeling. Then, we discuss methods developed and opportunities for modeling disordered regions and leveraging *in situ* data. Finally, we end with an outlook summarizing other open problems in integrative modeling.

## Integrative modeling methods

Several methods have been developed for integrative structure determination (Table 1). A subset of these including Integrative Modeling Platform (IMP), High Ambiguity Driven DOCKing (HADDOCK), and Assembline (Alber et al., 2007; Dominguez, Boelens, & Bonvin, 2003; Honorato et al., 2024; Rantos, Karius, & Kosinski, 2022; Russel et al., 2012) are discussed here. IMP is a framework for Bayesian integrative modeling that facilitates structure determination of macromolecular ensembles at multiple resolutions (multi-scale) and multiple states (multi-state) (Alber et al., 2007; Russel et al., 2012). A wide array of experimental data can be combined using IMP, for example in vivo genetic interactions, co-immunoprecipitation, FRET (Förster Resonance Energy Transfer), SAXS (small angle X-ray scattering), XLMS (chemical crosslinks from mass spectrometry), density maps from cryo electron-microscopy, and atomic structures from X-ray crystallography, NMR (Nuclear Magnetic Resonance), and AI-based predictions (Rout & Sali, 2019; Sali, 2021). The Bayesian inference framework allows for data from multiple sources to be integrated while considering the uncertainty in the data (Schneidman-Duhovny, Pellarin, & Sali, 2014). The modular design of IMP facilitates the mixing and matching of scoring functions and sampling algorithms. It has been used in the modeling of several large assemblies, most notably the nuclear pore complex (Akey et al., 2022; Alber et al., 2007; Rout & Sali, 2019; Sali, 2021; Singh et al., 2024). Recent advancements in IMP include Bayesian scoring functions for *in vivo* genetic interactions (Braberg et al., 2020), Bayesian model selection for optimizing model representation (Arvindekar, Pathak, Majila, & Viswanath, 2024), automated choice of sampling parameters (Pasani & Viswanath, 2021), and annotation of precision for model regions (Ullanat, Kasukurthi, & Viswanath, 2022).

Assembline is a protocol for integrative modeling that builds upon IMP, combining Xlink Analyzer, UCSF Chimera, and IMP to model large assemblies (Rantos et al., 2022). It is applicable for systems for which medium-resolution EM maps and a large number of atomic structures of subunits are available. It improves upon IMP by using pre-computed rigid body fits to EM maps to make the sampling more efficient. HADDOCK is a method for atomistic integrative modeling of protein complexes (Dominguez et al., 2003; Honorato et al., 2024). Experimental data from NMR, SAXS, XLMS, and mutagenesis studies are encoded as Ambiguous Interaction Restraints (AIR). Recent improvements to HADDOCK include the ability to model complexes of up to 20 macromolecules, new restraints based on cryo-EM maps, coarse-grained representations for efficient sampling, customizable pre- and post-processing steps, and a user-friendly web server for integrative modeling (Honorato et al., 2024).

Other than these, several methods allow fitting known protein structures into medium to low-resolution density maps, including MDFF and TEMPy-REFF (Beton, Mulvaney, Cragnolini, & Topf, 2024; Trabuco, Villa, Mitra, Frank, & Schulten, 2008). MDFF (Molecular dynamics flexible fitting) utilizes MD simulations for fitting structures into density maps by biasing the simulation using an additional potential derived from the density map (Trabuco et al., 2008). TEMPy-REFF (Responsibility-based Flexible-Fitting) refines an initial structure within a density map iteratively using the Expectation-Maximization algorithm (Beton et al., 2024).

## Recent examples in integrative modeling: focus on nuclear and cell adhesion complexes

Integrative modeling has shed light on diverse cellular processes by determining the structures of assemblies associated with them. A list of representative recently characterized integrative structures is presented (Table 2). Here, we discuss examples of recent integrative structural biology studies in nuclear trafficking, gene expression regulation, and cell–cell adhesion. These studies not only provide novel insights into the structure and function of these assemblies but also highlight areas for future applications and method development.

The nuclear pore complex (NPC) is a large macromolecular assembly in the nuclear envelope that connects the nucleus and cytoplasm and plays an important role in nuclear trafficking (Akey et al., 2022; Alber et al., 2007). Several recent studies have improved our understanding of the components of the NPC (Bley et al., 2022; Fontana et al., 2022; Singh et al., 2024; Yu et al., 2023; Zhu et al., 2022). Some of these studies involve the fitting of AlphaFold and experimentally determined structures into medium-resolution cryo-EM maps and cryo-electron tomograms (Bley et al., 2022; Fontana et al., 2022; Petrovic et al., 2022; Zhu et al., 2022). Other studies additionally incorporate biochemical data including chemical crosslinks (Singh et al., 2024). Together these studies have been used to characterize the structures of the cytoplasmic face, cytoplasmic ring, the linker-scaffold network, and the nuclear basket of the NPC. The resulting structures enabled the identification of

**Table 1.** Integrative modeling software

| Software | Authors | Reference | URL |
|---|---|---|---|
| ISD | Rieping, Habeck, & Nilges, (2005) | Rieping et al. (2005) | N/A |
| IMP | Russel et al. (2012) | Russel et al. (2012) | integrativemodeling.org |
| HADDOCK | Dominguez et al. (2003), Honorato et al. (2024) | Dominguez et al. (2003), Honorato et al. (2024) | rascar.science.uu.nl/haddock2.4 |
| Assembline | Rantos et al. (2022) | Rantos et al. (2022) | embl-hamburg.de/Assembline/ |
| PLUMED-ISDB | Bonomi & Camilloni, (2017) | Bonomi & Camilloni, (2017) | plumed.org |
| BioEn | Köfinger et al. (2019) | Köfinger et al. (2019) | github.com/bio-phys/BioEn |
| Rosetta | Simons, Kooperberg, Huang, & Baker, (1997), Leman et al. (2020) | Leman et al., (2020), Simons et al. (1997) | rosettacommons.org |
| CombFold | Shor & Schneidman-Duhovny, (2024b) | Shor & Schneidman-Duhovny, (2024a) | github.com/dina-lab3D/CombFold |
| CombDock | Inbar, Benyamini, Nussinov, & Wolfson, (2005), Schneidman-Duhovny & Wolfson, (2020) | Inbar et al. (2005), Schneidman-Duhovny & Wolfson, (2020) | bioinfo3d.cs.tau.ac.il/CombDock/download/ |

A list of commonly used integrative modeling software for large protein complexes. Each of these combines information from three or more experimental and/or computational sources. For a comprehensive overview, see (Bonomi et al., 2017; Habeck, 2023; Rout & Sali, 2019)

**Table 2.** A table summarizing a representative subset of recent integrative modeling studies

| Macromolecular assembly | Subcellular location | Software for integrative modeling | Data used | Authors and year |
|---|---|---|---|---|
| A3G-CRL5-Vif complex | Nucleus | IMP | Data from XLMS, and structures from X-ray crystallography and solution NMR | Kaake et al. (2021) |
| Apo-GAFab complex | Plasma membrane | IMP | Data from XLMS, and structures from X-ray crystallography | Gupta et al. (2020) |
| Bovine adenylyl cyclase 8 in complex with the G protein heterodimer | Plasma membrane | HADDOCK | Data from XLMS, and structures from X-ray crystallography and cryo-EM maps | Khanppnavar, B (2024) |
| CLOCK-BMAL1 bound to a nucleosome | Nucleus | Rosetta | Data from XLMS, and structures from X-ray crystallography and cryo-EM maps | Michael et al. (2023) |
| Desmosomal outer dense plaque | Plasma membrane | IMP | Data from cryo-ET, immuno-EM, yeast two-hybrid experiments, co-immunoprecipitation, in vitro overlay, in vivo co-localization assays, in silico sequence-based predictions of transmembrane and disordered regions, and structures from X-ray crystallography and homology modeling | Pasani et al. (2024) |
| Doublecortin-microtubule complex | Cytoplasm | IMP | Data from cryo-EM and XLMS, and structures obtained from cryo-EM | Rafiei et al. (2022) |
| gammaTuSC-Spc110 dimer complex | Nuclear membrane | IMP | Data from XLMS, and structures from cryo-EM maps | Brilot et al. (2021) |
| Human LINE–1 ORF2p | Nucleus | IMP | Data from cryo-EM and XLMS | Baldwin et al. (2024) |
| Intraflagellar transport - A (IFT-A) complex | Flagella | IMP | Data from XLMS cryo-ET, and AlphaFold structure predictions | McCafferty et al. (2022) |
| Mis18 Complex Assembly | Centromere | CombDock | Data from NS-EM and XLMS, and structures from X-ray crystallography | Thamkachy et al. (2024) |
| Mycobacterial ESX–5 type VII secretion system pore complex | Plasma membrane | IMP | Data from cryo-EM and XLMS, and structures obtained using X-ray crystallography and homology modeling | Beckham et al. (2021) |

**Table 2** *Continued*

| Macromolecular assembly | Subcellular location | Software for integrative modeling | Data used | Authors and year |
|---|---|---|---|---|
| Nexin-dynein regulatory complex | Cilia | Assembline | Data from XLMS and Alphafold structure predictions. | Ghanaeian et al. (2023) |
| Nuclear Basket of NPC | Nuclear membrane | IMP | Data from quantitative mass spectrometry, XLMS, cryo-ET, immuno-EM, biochemical studies, and bioinformatics predictions, and prior integrative models | Singh et al. (2024) |
| Nuclear Pore Complex (NPC) | Nuclear membrane | IMP | Data from cryo-ET, cryo-EM, XLMS, quantitative fluorescence imaging, and biochemical studies, and Alphafold structure predictions | Akey et al. (2023) |
| NuRD subcomplexes | Nucleus | IMP | Data from SEC-MALLS, DIA-MS, XLMS, negative-stain EM, and structures from X-ray crystallography, NMR spectroscopy, secondary structure predictions, and homology models | Arvindekar et al. (2022) |
| SARS-CoV2 Nsp1, Nsp2 and nucleocapsid proteins | Host cytoplasm and viral nucleocapsid | CombDock | Data from XLMS and structures from AlphaFold2 and homology modeling | Slavin et al. (2021) |
| SMC5/6 complex | Nucleus | IMP | Data from NS-EM and XLMS, and structures obtained using X-ray crystallography, cryo-EM, comparative modeling, and coiled-coil predictions | Yu et al. (2021) |
| Transglutaminase 2 in complex with plasma fibronectin type III modules 14 and 15 | Extracellular matrix | HADDOCK | Data from XLMS, and structures from X-ray crystallography | Selcuk et al. (2024) |
| Type III Secretion System | Plasma membrane, cell wall | Assembline | Data from XLMS and structures from cryo-EM and NMR spectroscopy | Flacht et al. (2023) |
| WDR76—SPIN1—nucleosome complex | Nucleus | HADDOCK, IMP | Data from XLMS and structures from X-ray crystallography and I-TASSER structure predictions | Liu et al. (2024) |

Abbreviations: DIA-MS, Data independent acquisition mass spectrometry; EM, Electron microscopy; ET, Electron tomography; NMR, Nuclear magnetic resonance; NS, Negative staining; SEC-MALLS, Size exclusion chromatography—multi-angle laser light scattering; XLMS, Crosslinking coupled with mass spectrometry.

novel interfaces between disordered nucleoporins (Nups) (Fontana et al., 2022; Zhu et al., 2022), elucidated the function of nucleoporins—Nup38 and the Cytoplasmic Filament Nucleoporin (CFNC) (Bley et al., 2022), delineated the role of Mlp/Trp in assisting mRNP transport (Bley et al., 2022; Fontana et al., 2022; Singh et al., 2024; Yu et al., 2023; Zhu et al., 2022), and revealed the plasticity and robustness of the inner ring (Petrovic et al., 2022). Finally, another study determined the distribution of intrinsically disordered nucleoporins in the NPC and their motion in the central channel using fluorescence lifetime imaging of fluorescence resonance energy transfer (FLIM-FRET) and coarse-grained molecular dynamic (MD) simulations (Yu et al., 2023).

Whereas the above studies are on components of the NPC, (Akey et al., 2022, 2023; Mosalaganti et al., 2022) determined comprehensive integrative structures of the entire NPC. These studies integrate *in situ* cryo-electron tomography data with AlphaFold or experimentally determined structures (Mosalaganti et al., 2022), and additionally cryo-EM maps, chemical crosslinks, and data from quantitative fluorescence imaging and biochemical studies to determine comprehensive structures of NPCs (Akey et al.,

2022, 2023). The structures revealed distinct dilated and constricted states of the complex and characterized the plasticity of the pore (Akey et al., 2022, 2023; Mosalaganti et al., 2022). Additionally, they localized precise anchoring sites for the intrinsically disordered Nups (Mosalaganti et al., 2022) and delineated the function of Pom153 in ring dilation (Akey et al., 2023).

The Nucleosome Remodeling and Deacetylase (NuRD) complex is a chromatin remodifying assembly that plays an important role in several cellular processes including transcriptional regulation, cell cycle progression, and cellular differentiation (Arvindekar et al., 2022). It consists of chromatin remodeling and deacetylase modules, connected by MBD and GATAD2 proteins. The structures of three subcomplexes of NuRD were determined by integrating data from negative-stain and low-resolution cryo-EM maps, X-ray crystallography, XLMS, SEC-MALS, DIA-MS, NMR spectroscopy, homology modeling, secondary structure predictions, and physical principles (Arvindekar et al., 2022). The integrative structures depict MBD in two states in NuRD and elucidate the role of the intrinsically disordered region of MBD in bridging the chromatin remodeling and deacetylase modules of NuRD.

Desmosomes are intercellular junctions that tether the inter-mediate filaments of adjacent cells in tissues under mechanical stress (Pasani, Menon, & Viswanath, 2024). The integrative structure of the desmosomal outer dense plaque (ODP) was determined by combining data from cryo-electron tomography, X-ray crystallography, immuno-electron microscopy, in vitro overlay, *in vivo* co-localization assays, Yeast Two-Hybrid (Y2H), co-immuno precipitation, *in-silico* sequence-based predictions of transmembrane and disordered regions, homology modeling, and stereochemistry (Pasani et al., 2024). The structure enabled the localization of disordered regions of Plakophilin (PKP) and Plakoglobin (PG) and the identification of novel protein–protein interfaces associated with them, leading to hypotheses about the functions of these disordered regions.

Two elements emerge as common across the aforementioned studies: they leverage *in situ* cryo-electron tomography data and the characterized systems contain significant fractions of disordered regions (Figure 1). This highlights two areas of immediate interest for method development: modeling with intrinsically disordered proteins (IDP) and utilizing data from cryo-electron tomography (cryo-TM), discussed in the following sections.

## Integrative modeling of intrinsically disordered proteins

Intrinsically disordered proteins (IDPs) are a class of proteins that lack a well-defined ordered structure in their monomeric state. Rather, they exist as an ensemble of interconverting conformers in equilibrium and hence are structurally heterogeneous (Baul et al., 2019; Lindorff-Larsen & Kragelund, 2021). This heterogeneity of IDPs also makes it challenging to characterize them both experimentally and computationally (Beck et al., 2024).

### Learning Representations for IDPs

Recently, protein language models (pLMs) have emerged as powerful tools for learning context-aware representations, providing a compact and informative approach to characterize the structural and functional properties of proteins (Bepler & Berger, 2021; Rives et al., 2021). pLMs enhance the performance of models on downstream tasks via transfer learning, eliminating the need to train a neural network from end to end. This approach is particularly beneficial while training models with small datasets.

Using pLMs for IDPs presents several challenges. First, pLMs trained only on sequences may not be able to capture the conformational heterogeneity of IDPs. Second, the databases used to train pLMs are dominated by ordered protein sequences, leading to a bias in the learned representations. Third, IDPs often function through transient interactions and context-dependent conformations, i.e., the same IDP may adopt different conformations with different binding partners. The state-of-the-art pLMs do not account for the environmental context and interacting partners and thus may not capture these transient interactions. Finally, the lack of structural data representative of IDP conformations poses a significant challenge in training models.

Advances in representation learning techniques are required for accurately characterizing the behavior of IDPs. Representations for IDPs could be improved by fine-tuning existing pLMs on IDP-specific tasks and/or by incorporating additional data on IDPs. Sequence alone might not be sufficient to capture the properties of IDPs; incorporating structural information or physics-based priors might allow pLMs to capture the complex

dynamics of IDPs (Wang, Wang, Evans, & Tiwary, 2024). Structure-aware pLMs have been recently developed (Peñaherrera & Koes, 2024; Sun & Shen, 2023; Wang et al., 2024). The same approach can be extended to IDPs. There is a need to obtain more structural data for IDPs (Jahn, Marquet, Heinzinger, & Rost, 2024). Whereas, experimental structural data remains important, acquiring it might be tedious and time-consuming. Computational approaches for generating realistic IDP conformational ensembles, such as MD simulations and generative models, would provide valuable experimental-like structural data. In the next section, we discuss methods for generating IDP ensembles.

## Generating IDP ensembles

Determining the conformational ensembles of IDPs is essential for understanding their functions. MD simulations are widely used for generating conformational ensembles. However, their reliability depends on the accuracy of force fields and the ergodicity of sampling (Bonomi, Heller, Camilloni, & Vendruscolo, 2017; Robustelli, Piana, & Shaw, 2018). Force fields typically used for folded proteins often fail to accurately capture the conformations of IDPs when compared with experimental data. Efforts for improving the force fields for IDPs focus on either refining the protein force field (Baul et al., 2019; Huang et al., 2017; Joseph et al., 2021), or accurately accounting for protein-water interactions (Best, Zheng, & Mittal, 2014; Nerenberg, Jo, So, Tripathy, & Head-Gordon, 2012; Robustelli et al., 2018; Vitalis & Pappu, 2009). Coarse-grained models that improve sampling by reducing the degrees of freedom have also been developed (Baratam & Srivastava, 2024; Baul et al., 2019; Joseph et al., 2021; Marrink, Risselada, Yefimov, Tieleman, & de Vries, 2007; Thomasen, Pesce, Roesgaard, Tesei, & Lindorff-Larsen, 2022).

Deep generative models offer a computationally efficient means for sampling conformations from a learned data distribution. Latent space embeddings from variational autoencoder (VAE) trained on IDP sequences (Mansoor, Baek, Park, Lee, & Baker, 2024), conditional generative adversarial networks (GAN) (Janson, Valdes-Garcia, Heo, & Feig, 2023), denoising diffusion probabilistic models (DDPM) (Janson & Feig, 2024; Zhu et al., 2024) have been used for generating all-atom and Cα coarse-grained ensembles of IDPs. More sophisticated approaches such as flow matching may also be employed for generating ensembles of IDPs. Notably, these aforementioned generative models leverage MD-generated ensembles for training.

Recent studies demonstrate the combined use of MD simulations and machine learning approaches to generate IDP conformers with the aim of predicting the biophysical properties of IDPs and designing IDP sequences (Lotthammer, Ginell, Griffith, Emenecker, & Holehouse, 2024; Pesce et al., 2024; Tesei et al., 2024). For example, the ALBATROSS deep learning model was developed for predicting the biophysical properties of IDPs, such as the radius of gyration, by training on IDP ensembles generated via the MPIPI-GG model (Lotthammer et al., 2024). Similarly, support vector regression models were trained to predict chain compaction for IDP sequences using IDP ensembles generated by the CALVADOS model (Tesei et al., 2024). Lastly, a method for designing IDP sequences with pre-defined conformational properties was developed by combining ensemble generation using CALVADOS with alchemical free-energy calculations within a Markov Chain Monte Carlo (MCMC) optimization framework (Pesce et al., 2024).

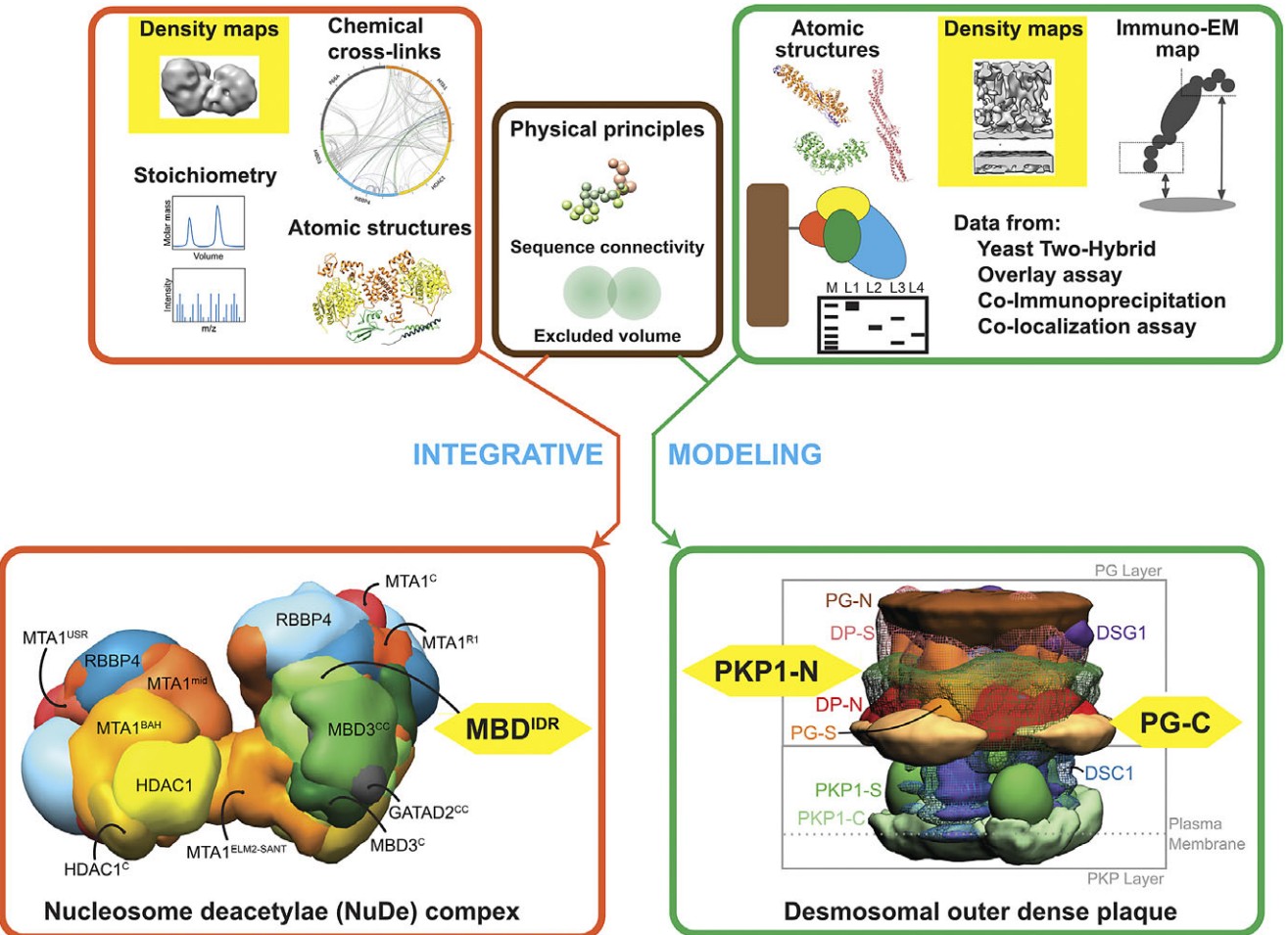

**Figure 1.** Frontiers in integrative structure determination. Schematic describing integrative structure determination for the nucleosome remodeling and deacetylase complex (orange box) and the desmosomal outer dense plaque (green box) combining data from multiple sources. Input low-resolution cryo-EM and cryo-ET maps and intrinsically disordered regions in both complexes are highlighted in yellow.

## Integrating experimental data for generating IDP ensembles

Broadly, experimental data can be utilized for modeling IDPs in several ways: validation of generated ensembles, reweighting generated ensembles using experimental data, incorporating experimental data as restraints for sampling conformations, or using experimental data to improve existing force fields (Bernetti & Bussi, 2023; Chan-Yao-Chong, Durand, & Ha-Duong, 2019; Fisher & Stultz, 2011). A comprehensive list of methods can be found in reviews on this topic (Bonomi et al., 2017; Habeck, 2023).

First, ensemble validation involves generating realistic ensembles of IDPs and validating the results with experimental data (Chan-Yao-Chong et al., 2019). Due to their ability to capture the dynamics of IDPs, NMR, and SAS data are most commonly used for validating the generated ensembles for IDPs (Baratam & Srivastava, 2024; Shrestha, Smith, & Petridis, 2021). Second, ensemble weighting involves using experimental data to refine an existing ensemble, to minimize deviation of the ensemble from the observed data (Chan-Yao-Chong et al., 2019). This can be achieved by maximum parsimony (SES Berlin et al., 2013) or maximum entropy (Pitera & Chodera, 2012; Roux & Weare, 2013; Cavalli, Camilloni, & Vendruscolo, 2013) (EROS Różycki, Kim, & Hummer, 2011, (BioEn Hummer & Köfinger, 2015), and ABSURD (Salvi, Abyzov, & Blackledge, 2016). Bayesian inference methods allow consideration of

uncertainty in data (Fisher, Ullman, & Stultz, 2013; Lincoff et al., 2020). Combining Bayesian inference and maximum entropy methods helps overcome the limitations of each (Crehuet, Buigues, Salvatella, & Lindorff-Larsen, 2019; Fröhlking, Bernetti, & Bussi, 2023). Deep learning models in combination with Bayesian and maximum entropy methods can also be used for refining an initial pool of conformations (DynamICE: Zhang, Haghighatlari, et al., 2023). Third, experimental data can also be used as restraints to guide simulations (Chan-Yao-Chong et al., 2019). Metainference uses Bayesian inference for incorporating noisy, ensemble-averaged experimental data using replica-averaged modeling (Bonomi, Camilloni, Cavalli, & Vendruscolo, 2016; Bonomi, Camilloni, & Vendruscolo, 2016). Similarly, parallel replica ensemble restraints based on SAXS data were used in MD simulations of IDPs (Hermann & Hub, 2019). Finally, experimental data can also be used for improving existing force fields on the fly using a Maximum Entropy approach (Cesari, Gil-Ley, & Bussi, 2016).

A holistic understanding of the dynamic behavior of IDPs requires realistic conformational ensembles that can be generated using MD simulations and deep generative models. MD simulations can provide experimental-like ensembles for training deep generative models; the latter may aid in improving force fields, enhancing sampling of IDP conformations, and analyzing the

ensemble generated via MD. Thus, an integrated approach would enable overcoming the limitations of each and improving our understanding of the dynamic nature of IDPs.

### Integrative structure determination using *in situ* data

Cryo-electron tomography (cryo-ET) is a cryo-EM imaging technique that enables structural characterization of macromolecular species (macromolecules, their complexes, and assemblies), in their native cellular environment at nanometer resolution (Gubins et al., 2020; Lamm et al., 2022). High-throughput localization and identification of macromolecular species within a tomogram can provide insights into their conformational heterogeneity, potential interactors, counts, and distributions within the cell (Arvindekar, Majila, & Viswanath, 2024; Beck et al., 2024; Förster, Han, & Beck, 2010; McCafferty et al., 2024). Integrating cryo-ET data along with complementary data from experiments such as XLMS, Y2H, cryo-EM Single Particle Analysis (SPA), FRET, AI-based structure predictions, and prior structural models can help build a comprehensive structural atlas of the cell (Beck et al., 2024; Förster et al., 2010; McCafferty et al., 2024). However, the intracellular crowding, compositional heterogeneity and low copy numbers of macromolecular species, the low signal-to-noise ratio, and the missing wedge in the tomography data pose significant challenges for localizing and identifying macromolecules in the tomograms (Moebel et al., 2021; Pyle & Zanetti, 2021).

### Localization and identification of macromolecular species with known structures

Macromolecular species with known structures are often annotated in tomograms either manually or by template matching. Manual particle annotation, however, is time-consuming, laborious, error-prone, and not suitable for high-throughput workflows (Lamm et al., 2022). Template matching involves using a low-pass filtered template of the known structure of a target macromolecule to localize similar densities in the tomogram (Frangakis et al., 2002). Methods for template matching are under active development (Cruz-León et al., 2024; Maurer, Siggel, & Kosinski, 2024). For example, the use of high-resolution information and template-specific search parameter optimization for objective, comprehensive, and high-confidence localization and identification of macromolecular species in tomograms was recently proposed (Cruz-León et al., 2024).

In addition to template matching, several supervised learning methods have also been recently developed. Two such deep learning-based methods, DeepFinder and DeePiCt, utilize convolutional neural networks (CNNs) for simultaneous localization and identification of macromolecular species (de Teresa-Trueba et al., 2023; Moebel et al., 2021). Another deep learning-based object detection method, MemBrain, was developed for estimating the localizations and orientations of membrane-embedded macromolecules (Lamm et al., 2022, 2024). These approaches have been shown to outperform template matching for localizing macromolecules (de Teresa-Trueba et al., 2023; Gubins et al., 2020; Lamm et al., 2022; Moebel et al., 2021). However, similar to manual annotation and template matching, these supervised learning approaches are limited to macromolecules with known structures. They are not suitable for high-throughput workflows and *de novo* structural characterization of macromolecular species (de Teresa-Trueba et al., 2023; Gubins et al., 2020; Lamm et al., 2022; Moebel et al., 2021).

### *de novo* localization and identification of species

For de novo structural characterization of macromolecular species with unknown structures, deep metric learning-based approaches, such as TomoTwin, and unsupervised learning approaches, such as Multi-Pattern Pursuit (MPP) and Deep Iterative Subtomogram Clustering Approach (DISCA) were recently developed (Rice et al., 2023; Xu et al., 2019; Zeng et al., 2023). These approaches aim to cluster subtomograms based on their structural similarity. Subtomogram averaging on the clustered subtomograms can aid in the structural characterization of macromolecular species at 10–20 Å resolutions (Rice et al., 2023; Zeng et al., 2023). These approaches are currently sensitive to noise in the tomograms and the size and abundance of the macromolecular species. However, they hold great promise for de novo high-throughput structural characterization of macromolecular species using tomographic data.

### Visual proteomics

Visual proteomics is an approach that aims to build molecular atlases that encapsulate structural descriptions of macromolecules within the cell using methods such as cryo-ET (Beck et al., 2024; Förster et al., 2010; McCafferty et al., 2024). This approach is inherently integrative. Given a tomogram, large macromolecular species with known atomic structures can be localized and identified within it using methods like template matching. Densities with unknown macromolecular identities can be obtained using the de novo approaches described above. The in situ structures of these uncharacterized macromolecular species can then be determined using an integrative approach by rigid fitting of structures obtained using cryo-EM SPA, X-ray crystallography, and AI-based structure predictions along with data from orthogonal experiments such as fluorescence microscopy and XLMS (Beck et al., 2024; Förster et al., 2010; McCafferty et al., 2024). For example, recent studies used integrative approaches to combine data from cryo-ET, SPA with cryo-EM, mass spectrometry, and predictions from AlphaFold to understand the molecular architecture of the human IFT-A and IFT-B complexes (Hesketh et al., 2022) and microtubule doublets in mouse sperm cells (Chen et al., 2023). In summary, utilizing cryo-ET data in an integrative approach can provide insights into interactors of a macromolecular species, associated protein communities, and larger cellular neighborhoods (Beck et al., 2024; Förster et al., 2010; McCafferty et al., 2024).

### Outlook

Integrative modeling has progressed significantly in the past decade, as evidenced by the increasing number, size, and precision of structures deposited to the PDB-Dev and integrated into the PDB (https://pdb-dev.wwpdb.org) (Saltzberg et al., 2021; Vallat et al., 2021). Integrative structural biology plays a crucial role in the era of AI-based structure predictions. Experimental data from rapidly advancing techniques such as cryo-electron tomography, and AI-based predictions can complement each other within an integrative framework (Arvindekar, Majila, & Viswanath, 2024; Beck et al., 2024; McCafferty et al., 2024; Shor & Schneidman-Duhovny, 2024b). This approach has proved powerful for several systems

such as ciliary complexes and nuclear pore complexes (Chen et al., 2023; Fontana et al., 2022; Hesketh et al., 2022; McCafferty et al., 2024; Mosalaganti et al., 2022; Zhu et al., 2022). Alphafold and similar AI-based prediction methods can increasingly solve structures for larger and more complex systems (Abramson et al., 2024). However, their applicability to solving entire structures of large assemblies remains an open question as they are limited by the GPU memory as well as the availability of training data. For example, membrane proteins and IDPs are under-represented in the training data (Carugo & Djinović-Carugo, 2023; Dobson et al., 2023). The low-pLDDT regions in Alphafold structures often coincide with IDRs, suggesting that Alphafold may be used to predict these regions (Wilson, Choy, & Karttunen, 2022). In contrast, in cases where Alphafold predicts structures of IDPs with high confidence, these regions typically represent the folded conformations of the IDPs, indicating a disorder-to-order transition in the presence of a partner (Alderson, Pritišanac, Kolarić, Moses, & Forman-Kay, 2023; Wilson et al., 2022). Nonetheless, the static structures from Alphafold are not an accurate representation of the dynamic behavior of IDPs, characterized by an ensemble of conformations (Ruff & Pappu, 2021).

In this Perspective, we highlighted two emerging frontiers for method development in integrative modeling: modeling disordered regions and modeling with data from cryo-electron tomography. Here, we briefly point to other open areas in integrative modeling that are the subject of current studies and/or may benefit from timely method development. First, a lack of knowledge about the system stoichiometry is one of the challenges for starting integrative modeling. Methods to estimate the stoichiometry based on the confidence of AI-based predictions are only beginning to be developed and are not yet generalizable (Chim & Elofsson, 2024; Shor & Schneidman-Duhovny, 2024b, 2024a). Second, methods for incorporating in vivo data in modeling are required. Recently, in vivo genetic interaction measurements were encoded as Bayesian distance restraints for integrative modeling of assemblies (Braberg et al., 2020). Similarly, methods for integrating other in vivo data such as data from super-resolution microscopy may also be developed to model larger cellular neighborhoods. Third, on the model representation front, it would be beneficial to determine system representation using objective measures instead of fixing them ad hoc (Arvindekar, Pathak, et al., 2024; Viswanath & Sali, 2019). Current methods for optimizing representations are limited to assessing a small number of candidate representations (Arvindekar, Pathak, et al., 2024; Viswanath & Sali, 2019). Methods that enable sampling and assessing a large number of representations, for example by dynamically varying the model representations during sampling, would benefit integrative modeling (Viswanath & Sali, 2019). Fourth, methods for integrative modeling of dynamic systems with multiple discrete states and/or a continuum of states are also continually advancing (Habeck, 2023; Hoff, Thomasen, Lindorff-Larsen, & Bonomi, 2024; Hoff, Zinke, Izadi-Pruneyre, & Bonomi, 2024; Lincoff et al., 2020; Potrzebowski, Trewhella, & Andre, 2018). Fifth, sampling procedures in integrative modeling may be improved by leveraging the recent advances in deep learning, particularly in generative modeling. Specifically, recent generative modeling methods for protein structure prediction may be extended to incorporate experimental data, potentially leading to more efficient sampling procedures than the current stochastic sampling methods (Jing, Berger, & Jaakkola, 2024; Watson et al., 2023; Wu et al., 2024; Zheng et al., 2024). Finally, methods for comprehensive validation of integrative models, including assessment of model uncertainty and Bayesian assessment of fit to different kinds of input data are

also necessary and are under development (Sali et al., 2015; Vallat et al., 2021). In all, these efforts will facilitate faster, more accurate, and more precise characterization of larger assemblies (Sali, 2021). The grand challenge in the field is to construct spatiotemporal models of entire cells. Integrative models of assemblies can contribute directly to this effort via metamodeling efforts that involve the integration of models at different scales to address the grand challenge (Raveh et al., 2021).

**Open peer review.** To view the open peer review materials for this article, please visit http://doi.org/10.1017/qrd.2024.15.

**Acknowledgments.** Molecular graphics images were produced using the UCSF Chimera and UCSF ChimeraX packages from the Resource for Biocomputing, Visualization, and Informatics at the University of California, San Francisco (supported by NIH P41 RR001081, NIH R01-GM129325, and National Institute of Allergy and Infectious Diseases).

**Author contribution.** K.M., S.A., and M.J.: reading and synthesis. K.M., S.A., M.J., and S.V.: writing: original draft, writing: revision. K.M.: visualization. S.V.: supervision, funding.

**Funding.** This work has been supported by the following grants: Department of Atomic Energy (DAE) TIFR grant RTI 4006, Department of Science and Technology (DST) SERB grant SPG/2020/000475, and Department of Biotechnology (DBT) BT/PR40323/BTIS/137/78/2023 from the Government of India to S.V.

**Competing interest.** None declared.

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
