## [Reviewer Report]

The review by Majila et al. presents a concise overview of the state-of-the-art in integrative structural biology, with a particular focus on the challenges in determining structural ensembles of disordered systems, like IDPs or IDRs, and utilising in situ data, such as cryo-electron tomography. This review is extremely timely and will be very informative for new researchers approaching the field. However, while I understand the difficulty of providing a comprehensive overview in the limited space available, I personally find that some key players and publications in the field are not sufficiently represented. In particular (and with the reassurance that none of the reference mentioned below is from this reviewer):

1) when discussing generation of IDPs ensembles with in silico approaches, the work by Kresten Lindorff-Larsen (in particular this: https://doi.org/10.1038/s41586-023-07004-5) and Alex Holehouse (https://doi.org/10.1038/s41592-023-02159-5) labs should be mentioned

2) when discussing integrative approaches for IDP ensembles determination, the work by the labs of Kresten Lindorff-Larsen, Gerhard Hummer, Teresa Head-Gordon, Giovanni Bussi, John Chodera, Martin Blackledge (as bare minimum), should be mentioned

Finally, I think that this review would be even more informative if two tables were added to the manuscript:

1) a Table summarising the software available for integrative structure determination, with minimal information, such as Authors, reference publication, URL;

2) a Table summarising some of the recent macromolecular complexes determined by integrative approaches, with informations such as Authors, reference publication, software used, data used. This table would make a more comprehensive overview compared to the few examples mentioned in the manuscript

---

## [Reviewer Report]

Kartik Majila et al. wrote a perspective about integrative structural modelling and integration of various experimental and computational methods of disordered proteins. The article is well written, it is an informative source and most citations are up to date. There are a few minor corrections the reviewer would like to highlight:

1) Since authors mostly discuss examples from nuclear trafficking, gene expression regulation, and cell-cell adhesion, the reviewer believes the title of the manuscript should be less general than it is at the present and relate to the examples mentioned. IDPs exist everywhere and it authors describe only fraction of this broad subject. Then the title of the paragraph “Recent advances of integrative structures” will be implicitly more specific.

2) Authors introduce IDPs only in the third paragraph, since this is one of the main topics of this perspective, can authors move this part to introduction and then elaborate more with specific examples?

3) Can authors elaborate more on Alphafold and its recent advances and limitations when it comes to IDPs? i) Int J Mol Sci. 2022 May; 23(9): 4591. ii) J Mol Biol, 2021 Oct 1;433(20):167208, iii) Proc Natl Acad Sci USA, 2023 Oct 31;120(44):e2304302120.

4) Molecular dynamics flexible fitting (MDFF) is a standard method of integrative modelling. In this method the additional bias allows to fit the macromolecular assemblies into low or high resolution experimentally derived density maps. Can authors introduce this technique with appropriate citation and examples?

5) “Coarse-grained models that improve sampling by reducing the degrees of freedom have also been developed (Baratam & Srivastava, 2024; Baul et al., 2019; Joseph et al., 2021)” – can authors add original Martini force field (ff) citation since this is the most popular and broadly used model used. This ff includes parameters for protein, lipids, carbohydrates, small molecules, polymers and many more.

---

## [Editor Report]

We have received two reviewers comments. They are both positive but suggested changes. Please have a look at it. Hope to receive your revised manuscript.

---

## [Reviewer Report]

Authors addressed all reviewer comments, this article is suitable for publication in the present form.